# Normalizing the permafrost carbon feedback contribution to the Transient Climate Response to Cumulative Carbon Emissions and the Zero Emissions Commitment

Norman J. Steinert<sup>1,2</sup> and Benjamin M. Sanderson<sup>1</sup>

<sup>1</sup>CICERO Center for International Climate Research, Oslo, Norway

<sup>2</sup>Potsdam Institute for Climate Impact Research (PIK), Member of the Leibniz Association, Potsdam, Germany

**Correspondence:** Norman J. Steinert (norman.steinert@cicero.oslo.no)

Abstract. As permafrost thaws, the permafrost carbon feedback (PCF) can amplify the Transient Climate Response to Cumulative Carbon Emissions (TCRE) and the Zero Emissions Commitment (ZEC) by introducing additional greenhouse gases into the atmosphere. Using a basic permafrost carbon response model coupled to the simple climate model FaIR, we estimate this feedback's contribution to TCRE and ZEC100 (ZEC at 100 years after emission cessation) and find that it can substantially increase estimates of these climate metrics. TCRE is increased by 0.12 % per PgCeq/°C of PCF and is robust in scenarios with various emission rates. ZEC100's increase is emission-rate dependent but is increased by 0.006 °C per PgCeq/°C of PCF for emission rates of 10 PgC/yr and is robust for varying emission rates when time-integrated warming is considered. Relating these climate metrics to permafrost carbon emissions allows the normalization of the PCF contribution to TCRE and ZEC by discounting its uncertainties.

### 10 1 Introduction

The Transient Climate Response to Cumulative Carbon Emissions (TCRE) measures the global surface temperature increase per 1000 PgC of carbon emitted, given a near-linear relationship seen in the majority of Earth system models (Liddicoat et al., 2021; Gillett et al., 2013), making it a key metric for estimating remaining carbon budgets and guiding climate policy (MacDougall, 2016). Equally relevant for the Earth system response to warming is the Zero Emissions Commitment (ZEC), which refers to the expected temperature change that occurs after anthropogenic emissions are completely stopped (Palazzo Corner et al., 2023).

Both TCRE and ZEC are influenced by biogeochemical and thermal climate feedbacks, which can either amplify (positive feedbacks) or dampen (negative feedbacks) warming during transient warming periods and after carbon emission stabilization. However, consistent modeling of TCRE and ZEC in future climate scenarios is hindered by inter-model discrepancies in representing processes governing the thermal climate response that stems from inter-model differences in the effect of physical climate feedbacks and planetary heat uptake (Williams et al., 2020). These processes lead to significant uncertainty and disagreement between different climate models regarding their TCRE (Williams et al., 2020) and the sign and magnitude of temperature change following the cessation of carbon emissions (MacDougall et al., 2020).

One such climate feedback is the permafrost carbon feedback (PCF; Schuur et al., 2022). Permafrost is permanently frozen ground that remains at or below 0°C for at least two consecutive years, typically found in polar regions. As temperatures rise, microbial decomposition from thawing permafrost emits additional carbon into the atmosphere in the form of CO<sub>2</sub> and methane (Schuur et al., 2022). Earlier assessments did not fully account for the PCF, potentially underestimating its effect on TCRE and ZEC (Canadell et al., 2021; Natali et al., 2021). However, this feedback can amplify warming and potentially increase TCRE and ZEC estimates. For ZEC, a perturbed parameter experiments conducted with an Earth system model of intermediate complexity that represents the PCF to climate change projects an additional 0.27°C warming 500 years post-emissions due to prolonged carbon release, with permafrost carbon loss partly offsetting mitigation efforts (MacDougall, 2021). There is still significant uncertainty in the quantification of the PCF due to structural differences between the models used to quantify the PCF, particularly in representing soil carbon decomposition response to climatic change (Burke et al., 2017; Canadell et al., 2021). This underscores the need to integrate the PCF (and propagate its uncertainties) into climate projections to refine TCRE and ZEC estimates.

Here, we quantify the contribution of the PCF to estimate TCRE and ZEC by using a basic permafrost carbon response model coupled to the simple climate model FaIR. This allows to obtain a relationship between permafrost carbon emissions and TCRE and ZEC, so that the contribution of permafrost carbon emissions to these climate metrics can be independently inferred from knowledge of the PCF alone. This quantification considers the response of currently-frozen carbon in the soil to warming, not including further positive and negative (localized) climate feedbacks, such as from vegetation interactions or its changing distribution (Pugh et al., 2018), nitrogen fertilization (Burke et al., 2022), or nonlinear dynamics of biogeochemical and biogeophysical processes (Nitzbon et al., 2024). However, a comprehensive sampling for climate and carbon response uncertainties constrained to the responses of the more complex models, although ambiguous, implicitly emulates some of these processes in our modeling approach.

### 45 2 Permafrost carbon response model: PerCX

35

To estimate the permafrost carbon feedback contribution to TCRE and ZEC, we compare two versions of the FaIR simple climate model: the standard version (v1.6.4; Smith et al., 2018), and a modified version that incorporates an idealized representation of the PCF (FaIR-PCF hereafter). To FaIR-PCF, we introduce a permafrost carbon response model (PerCX) described in Eqs. 1–3. The carbon response to climate in PerCX is determined by the sum of the CO<sub>2</sub> and CH<sub>4</sub> responses:

$$50 \quad \Delta C_{PF}(t) = \Delta C_{CO_2}(t) + \Delta C_{CH_4}(t). \tag{1}$$

Both  $\Delta C_{CO_2}(t)$  and  $\Delta C_{CH_4}(t)$ , derived as carbon emissions, are determined by an exponential decay function taking into account the temperature history:

$$\Delta C_{CO_2}(t) = \int_0^t A_{CO_2} \cdot \Delta T(t') \cdot \frac{C_p(t')}{C_p(0)} \cdot e^{\frac{-(t-t')}{\tau}} dt', \tag{2}$$

and

75

55 
$$\Delta C_{CH_4}(t) = \int_{0}^{t} A_{CH_4} \cdot \Delta T(t') \cdot \frac{C_p(t')}{C_p(0)} \cdot e^{\frac{-(t-t')}{\tau}} dt',$$
 (3)

with  $C_p(0)$  denoting the initial carbon pool at time t=0 and  $C_p(t)$  denoting the time-evolving combined leftover  $\mathrm{CO}_2$  and  $\mathrm{CH}_4$  carbon pool after some emissions have been caused, i.e.,  $C_p(t) = C_p(0) - \int_0^t \Delta C_{PF}(t) dt$ . Additionally, there are currently three degrees of freedom in PerCX: the response amplitudes  $A_{CO_2}$  for  $\mathrm{CO}_2$  and  $A_{CH_4}$  for  $\mathrm{CH}_4$ , and a combined response timescale  $\tau$ . Here, we assume the response timescales for  $\mathrm{CO}_2$  and  $\mathrm{CH}_4$  to be the same, effectively releasing carbon at an identical rate. Note that this only refers to the rate of emissions, not the individual climate effects of  $\mathrm{CO}_2$  and  $\mathrm{CH}_4$ , which is considered by FaIR internally. We also note that under circumstances,  $\mathrm{CH}_4$  might be more volatile, for example, when abrupt thaw processes cause nonlinear  $\mathrm{CH}_4$  responses to warming (Turetsky et al., 2020) - a scenario that is currently not captured in  $\mathrm{PerCX}$ .

 $A_{CO_2},\ A_{CH_4}$ , and au are calibrated by randomly sampling 1000 combinations of these parameters using a uniform independent distribution. The sampling range per parameter is chosen so that it exceeds the range (upper and lower end) of CMIP-model-based permafrost carbon loss under historical + SSP2-4.5 scenario warming (Fig. 1a) of 4–48 PgC/°C at 2100 (Fig. 1b) given in the The Intergovernmental Panel on Climate Change's (IPCC) 6th assessment report (AR6, Canadell et al., 2021, Box 5.1;). Then, only those parameter combinations are kept that give carbon loss estimates that fall into the AR6 range (here 775 of 1000 combinations). Note that this accounts for varying shapes of exponential decay, as long its response conforms to the constraining range at 2100. Additionally, the initial carbon pool size is set to  $C_p(0)$ =1400 PgC (Meredith et al., 2022; Schuur et al., 2022) and a CO<sub>2</sub>-to-CH<sub>4</sub> ratio of 6/1 is enforced so that  $C_{CO_2} = \frac{6}{7}C_p$  and  $C_{CH_4} = \frac{1}{7}C_p$ . Note that this constrained ensemble (Fig. 1b) still constitutes a big uncertainty reflected in idealized temperature 'Transient' ramp-up of 1.5°C per century and 'Stabilization' at 3°C scenario simulations (Fig. 1c,d), where PerCX's carbon loss ranges between 118–728 and 146–828 PgCeq after 400 simulation years, with medians of 396 and 480 PgCeq, respectively.

Taking the combined CO<sub>2</sub> (in units PgC) and CH<sub>4</sub> (converted to units Mt) emissions as input, FaIR simulates their individual effects on the climate, including a consideration of their respective lifetimes. From a pool of 2237 parameter combinations representing climate sensitivity uncertainty in FaIR (Forster et al., 2021; Smith et al., 2021), and the pool of 775 coefficient combinations from PerCX - representing carbon loss sensitivity, we perform uniform independent sampling for 1000 parameter combinations from these two pools to explore modeling uncertainties. The results of FaIR-PCF are tested against the UVic intermediate complexity Earth System Model (MacDougall, 2021) in Appendix A (Fig. A1).

Figure 1. a) Global mean temperature anomaly ( $\Delta$ GMST) of 29 CMIP6 Earth system models for the *historical* + *SSP2-4.5* scenario relative to pre-industrial conditions. b) Carbon loss response in PerCX to the CMIP6 ensemble mean in panel a to derive a prior sample (gray). CMIP-model-based permafrost carbon loss estimates from IPCC AR6 are then used to constrain PerCX parameter combinations (teal). c) Idealized temperature 'Transient' ramp-up and 'Stabilization' scenarios that are used to illustrate in d) the out-of-sample response of PerCX to these idealized scenarios.

### 3 Results

Figure 2a shows results for simulations following the flat10MIP protocol for the *flat10* experiment (Sanderson et al., 2024), assuming 10 PgC/year emitted constantly over 200 years, which allows a quantification of TCRE. Here, TCRE and PCF are quantified over 1000 PgC of cumulative emissions (gray shading). For the standard FaIR, the mean temperature response exhibits a TCRE of 1.39 (1.25–1.52) °C/EgC (blue; median with minimum-to-maximum estimates in brackets) - on the lower end of the range of CMIP models (Arora et al., 2020). Considering the permafrost carbon response in FaIR-PCF yields a TCRE of 1.45 (1.32–1.59) °C/EgC (orange) - a median increase by 0.06 °C/EgC, or 4.3 %. The effect of PCF is slightly increased

when calculating TCRE over 2000 PgC of cumulative emissions, with a standard TCRE of 1.35 (1.15–1.51) °C/EgC, increased by roughly 6.6 % due to PCF to 1.43 (1.21–1.61) °C/EgC.





Given the uncertainties in existing permafrost carbon loss simulations, we quantify the percentage change of TCRE due to PCF to permafrost carbon emissions for all combinations of our uncertainty sampling (n=1000). The PCF is 14 (2–29) PgCeq/°C and its impacts on TCRE ranges between 0–4.5 % (Fig. 2b). The emerging relationship yields a 0.12 % increase in TCRE per PgCeq/°C of PCF. Because the temperature response to cumulative emissions holds under various *flat10*-like scenarios with different emission rates, this relationship is also robust across variations of emission rates ranging from 5 to 40 PgC/year (Figs. A2, A3a–e, A4a). Therefore, the quantification of this relationship allows for a more generalized translation between TCRE and PCF under a range of PCF feedback strengths. Using this framework then also allows to infer a theoretical TCRE increase due to PCF if only the PCF is known. Or vice versa, quantifying TCRE differences allows to infer a quantification of the PCF in a given climate model.

A previous estimate by MacDougall and Friedlingstein (2015) using the UVic Earth System Climate Model shows a much higher base TCRE of 1.9 K/EgC - at the upper end of CMIP6 models, and a strong increase of roughly 16 % in TCRE due to PCF to 2.3 K/EgC. MacDougall and Friedlingstein (2015) acknowledge that UVic has one of the largest carbon releases from permafrost soils of any land surface model at the time of publication. They also report values for effective TCRE (i.e., including non-CO<sub>2</sub> forcers), which tend to be higher than CO<sub>2</sub>-only TCRE values. Still, UVic's increased TCRE values compared to FaIR indicate a larger climate sensitivity of UVic, which could explain some of the differences to the results presented here. Using our generalized approach allows to infer a substantially larger PCF in UVic (~134 PgC/°C), which is likely overestimated slightly due the difference between CO<sub>2</sub>-only TCRE and effective TCRE.

Equally, for the quantification of the PCF effect on ZEC, Figure 2c shows results for the *flat10-zec* experiment (Sanderson et al., 2024), following 10 PgC annual emissions for the first 100 years before emissions cease. The standard FaIR's ZEC100 (temperature 100 years after emissions cease relative to the year at which emissions cease) is 0.06 (-0.03–0.12) °C, whereas it is 0.14 (-0.01-0.33) °C for FaIR-PCF - a median increase of 0.08°C. This indicates that the PCF contribution to ZEC100 is more than half as large as the ZEC100 uncertainty range in the standard FaIR. These numbers are comparable with previous studies employing a similar simulation setup quantifying the PCF to add 0.09°C (0.04–0.21) °C to ZEC1000 after emitting 1000 PgC of CO<sub>2</sub> with an additional 0.04°C (0 to 0.06°C) arising from thaw-lagged permafrost thaw caused by rapid emission rates in standardized ZEC experiments (MacDougall, 2021).

Similar to TCRE, we further quantify ZEC100 relative to permafrost carbon emissions, which gives an increased by 0.006 °C per PgCeq/°C of PCF for emission rates of 10 PgC/yr (Fig. 2d). However, ZEC100's increase is emission-rate dependent (also see Fig. A2, A3), so that for smaller and larger emission rates ranging 5–40 PgC/yr, ZEC100's increase due to PCF varies between 0.003 and 0.05 °C per PgCeq/°C (Figs. A3,A4). This is due to the shortened ramp-up period allowing permafrost carbon emissions when emission rates are high, and vice versa. However, ZEC100's increase due to PCF is consistent across emission rates when the time-integrated temperature exposure is considered (Fig. A4d). This is also consistent with (MacDougall, 2021), who finds the PCF's relative impact to remain consistent across emission scenarios (1000 vs. 2000 PgC), though absolute carbon releases scale with total emissions. While the range of these feedbacks slightly increases ZEC, it doesn't fundamentally

Figure 2. a) Global mean surface temperature ( $\Delta$ GMST) relative to pre-industrial conditions vs cumulative carbon emissions of the *flat10* scenario of 10 PgC annual emissions for 200 years, without (blue) and with (orange) permafrost carbon feedback (median of parameter ensemble distribution; shading shows the min-to-max range). The gray lines denote CMIP6 models (top to bottom: ACCESS-ESM1-5, CESM2, MPI-ESM1-2-LR, GISS, GFDL-ESM4, NorESM2-LM) that have performed the same experimental design within flat10MIP (Sanderson et al., 2024)). For reference, the dashed and dotted black lines show previous estimates from MacDougall and Friedlingstein (2015). b) TCRE increase [%] due to permafrost carbon feedback versus permafrost carbon emissions (in PgCeq as the sum of CO<sub>2</sub> and CH<sub>4</sub> emissions) per degree of warming [°C]. Note that the uncertainty sampling is only shown for *flat10*, whereas *flat10*-like variations of that experiment constituting different warming rates (i.e., 5, 8, 20, and 40 PgC/yr) are shown as colored lines. c)  $\Delta$ GMST temporal evolution of the *flat10-zec* scenario, 10 PgC annual emissions for the first 100 years before emissions cease. ZEC100 is estimated as the difference between simulations years 200 and 100. d) Same as b) but for ZEC100 changes due to PCF [K]. The sample size for the parameter ensemble in panels b and d is n=1000. Again, panel d also show estimates for *flat10*-like scenarios with different emission rates.

alter the conclusion that ZEC remains near zero on inter-decadal scales after emissions cease. However, it becomes increasingly significant over centuries due to persistent carbon release from thawed soils under the elevated stabilization temperature (McGuire et al., 2018).

## 4 Conclusions



The relationships found here generalize the contribution of permafrost carbon emissions to TCRE and ZEC100, so that uncertainties in the strength of the PCF are accounted for. Hence, TCRE and ZEC100 differences due to PCF can be quantified as long as the PCF and the scenario temperature trajectory (i.e., time-integrated warming) are known. Still, uncertainty in TCRE and ZEC estimates remain, as the amount and rate of permafrost carbon release generally depend on several factors, including regional warming patterns, soil moisture, and microbial activity. However, regardless of whether current-generation climate models show a particularly weak or strong PCF, the relationship framework presented here allows to simply infer the PCF's contribution to TCRE and ZEC100 by using estimates provided here as a scaling-factor (for TCRE) and an addition (for ZEC).

As a caveat, permafrost emissions could continue contributing to atmospheric CO<sub>2</sub> and CH<sub>4</sub> long after anthropogenic emissions peak and could therefore increase the current ceiling of PCF estimates, specifically when long time scales (e.g., centuries to millennia) or non-linear responses to climate change are considered. Further, model specific results indicating that increased feedback strength could lead to non-linearity in TCRE (MacDougall and Friedlingstein, 2015) are not quantified with the current coupling of PercX and FaIR as used here. We therefore call for additional efforts and more complex models, e.g, permafrost-process based models, including Earth system models, to further explore the possibility for deviations from a linear TCRE relationship and modifications of ZEC. These results highlight the necessity of incorporating the permafrost carbon feedback into climate projections to avoid underestimating future warming and refining carbon budget assessments.

# Appendix A





The Appendices includes Figure A1–A4:

Figure A1 shows the comparison of PerCX to the UVic intermediate complexity model (MacDougall, 2021) for the esm-1pctbrch-1000PgC scenario, with additional PerCX results for the flat10-zec scenario for reference. The esm-1pct-brch-1000PgC scenario follows a 1% increase of emissions per year, until emissions are reduced to zero when 1000 PgC of cumulative emissions are reached. Since this scenario is CO<sub>2</sub>-concentration driven, these emissions are inferred from the default FaIR version 1.6.4 using the *fair.inverse* functionality. As noted by MacDougall (2021), equal in the application with UVic, this means that not all different model setups (i.e., parameter combinations or model variants) follow the 1% pathway exactly, but this treatment ensures a simplification of the data handling and analysis of the results and likely presents a negligible deviation of the results from the true 1% pathway of each model setups. UVic's median response to this scenario is a rapid loss of permafrost carbon over the first 100–200 years, where it start to slow down for several centuries before it slowly start to accelerate emission again towards the end of the simulation. In contrast, PerCX is slower in its initial median response to the forcing and starts to accelerate permafrost carbon loss when UVic starts to slow down before PerCX begins to stabilize towards the end of the simulation. At least in PerCX, the response differences between the esm-1pct-brch-1000PgC and flat10 scenarios are negligible. Together with the somewhat different temporal dynamic of the response between the two models, PerCX results in about 50% larger median cumulative permafrost carbon loss at the end of the simulation. However, the lower range (5th percentile) of both models is quite similar, while PerCX upper range (95th percentile) produces significantly larger carbon loss. This indicates that, in this model comparison, the median response of PerCX is biased by the upper end of the response range it is calibrated to. Since we here opted for calibration to a response range of CMIP models, and not a single model (which was in fact also tested during the calibration process), there are obvious differences between PerCX and UVic. Hence, the model differences are partly also an artifact of where UVic's permafrost carbon response lies in that AR6 range and how its median response differs from that of the average CMIP6 models. We note that, despite their differences, the functional form of the responses of PerCX and UVic are quite similar and it would be possible to calibrate PerCX to output from UVic, which would result in a much closer match between the two model responses.

Figure A2 shows the *flat10*-like and *flat10-zec*-like scenario response of temperature and permafrost carbon loss from FaIR-PCF versus time and cumulative emissions for various emission rates.

Figures A3 and A4 show the fits of TCRE and ZEC100 changes due to the permafrost carbon feedback for for various emission rates, where Figure A4 summarizes the results, also showing permafrost carbon feedback by time-integrated temperature exposure.

**Figure A1.** a) CO<sub>2</sub> emissions over time for the 1% increase scenario where emissions are abruptly decreased to zero (*esm-1pct-brch-1000PgC*) and the *flat10-zec* scenario. The emissions for the (*esm-1pct-brch-1000PgC*) are inferred from the default FaIR version 1.6.4, given that this scenario is CO<sub>2</sub>-concentration driven. b) Cumulative permafrost carbon loss for the UVic model (MacDougall, 2021) and PerCX for the *esm-1pct-brch-1000PgC* scenario. Additionally, PerCX results for the *flat10-zec* scenario are shown. Note that PerCX results are only showing the permafrost carbon emissions of the CO<sub>2</sub> component to allow for a direct comparison to UVic. The shaded areas denote the 5th–95th percentile ranges.

Figure A2. a) Global mean surface temperature ( $\Delta$ GMST) relative to pre-industrial conditions over time from FaIR-PCF for *flat10*-like scenarios with emission rates ranging 5–40 PgC/yr. b) Cumulative permafrost carbon loss over time for the same scenarios. c) and d) same as panels a and b but versus vs cumulative carbon emissions. e–f) same as panels a–d but for *flat10*-like scenario with the same variations in emission rates.

Figure A3. a–e) Changes in TCRE (ΔTCRE) by cumulative changes of the permafrost carbon feedback for *flat10*-like scenarios with emission rates ranging 5–40 PgC/yr. Note that despite the PCF being smaller, the higher the emission rates, the changes of TCRE change due to PCF are consistent across emission rates. f–j) Changes in ZEC100 by cumulative changes of the permafrost carbon feedback for the same scenarios. Here, the absolute change of ZEC100 is emission-rate dependent, increasing with increasing emission rates. Note that the results of *flat10-zec* in panels c and h are what is shown in Figure 2.

Figure A4. a) Changes in TCRE ( $\Delta$ TCRE) by cumulative changes of the permafrost carbon feedback for *flat10*-like scenarios with emission rates ranging 5–40 PgC/yr. b) same as a but for permafrost carbon feedback by time-integrated temperature exposure. Due to the shorter period until 1000 PgC of cumulative emissions for scenarios with higher emission rates, the time-integrated temperature exposure is less, permafrost carbon emissions are less, and vice versa. c and d) same as a and b but for ZEC100. Note that ZEC100 is emission-rate dependent for changes due to PCF but is consistent across emission rates when the time-integrated temperature exposure is considered when assessing the contribution of permafrost carbon emissions in panel d. Panels a and c replicate the results of Figure 2b,d.

Code availability. All relevant code information is either given within this manuscript or referenced accordingly.

Data availability. All relevant data information is either given within this manuscript or referenced accordingly.

| $Author\ contributions.$ | N.J.S and B.M.S des    | igned and directed the  | concept of this manu    | uscript. N.J.S co | nducted the model | development, run |
|--------------------------|------------------------|-------------------------|-------------------------|-------------------|-------------------|------------------|
| the model experiments    | s, performed the calcu | lations and wrote the m | nanuscript, all with re | evisions from B.  | M.S.              |                  |

Competing interests. The authors declare no competing interests.

Acknowledgements. This research has been supported by the Research Council of Norway through the project TRIFECTA (grant no. 334811).

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
