# Peer review of "Normalizing the permafrost carbon feedback contribution to the Transient Climate Response to Cumulative Carbon Emissions and the Zero Emissions Commitment"

_EGUsphere, 2025_

## Author Comment (AC2)

**Response to the reviewers' comments: EGUSPHERE-2025-1714. Normalizing the permafrost carbon feedback contribution to TCRE and ZEC**

The authors would like to thank the reviewers Chris Jones and Andrew MacDougall for their constructive suggestions and the time they devoted to reviewing the manuscript. We appreciate their contribution and tried to integrate and consider all suggestions. We think that the manuscript has improved with them.

The next sections contain a point by point response to the reviewers comments. Comments are labeled by reviewers and order of appearance, i.e. R1C2 is the second comment of reviewer 1.

**Reviewer #1 (Chris Jones)**

R1C1: *REVIEWER'S COMMENT:*

*Can you comment on the uncertainty due to the assumed CO2:CH4 ratio? Burke et al (http://www.the-cryosphere.net/6/1063/2012/tc-6-1063-2012.html) considered this and the spread in that study was quite wide. Why do you choose a 6:1 ratio? (line 62)*

AUTHORS' RESPONSE:

This ratio stems from the constraining process as it reflects the 6/1 ratio of reported values for the permafrost carbon feedback from the IPCC's 6th assessment report (AR6, Box 5.1; Canadell et al., 2021) from $CO_2$ (18 PgC °C–1) and $CH_4$ (2.8 PgCeq °C–1), based on a wide range of scenarios evaluated at 2100. This ratio is also fairly consistent considering the upper and lower bounds of the respective 5th–95th percentile range given in the report. We note that PerCX is calibrated carbon to equivalent emissions of $CH_4$, which makes it more consistent to the reported values from AR6. This also means that, in this paper, we assume the ratio of emissions between $CO_2$ and $CH_4$ to be constant over time. However, some of the influence of varying ratios should be implicitly captured by the range of responses reported in AR6. This uncertainty range by itself is already quite large, and an additional degree of freedom would potentially modify PerCX's response to also fall somewhere in this range. Regardless, we acknowledge that their ratio constitutes a potentially important source of uncertainty, and that it may also vary over time depending on local climate feedbacks (which is also not covered by the 2100 reference of the permafrost carbon feedback estimates in AR6). Neither are contained in the current version of PerCX (as referred to in line 53–54 of the

25      original manuscript), but we expect this source of uncertainty to be subject to exploration in future applications of PerCX.

R1C2: *REVIEWER'S COMMENT:*

*Related – in the same way that anthropogenic CO2 and CH4 have different relative contributions over time, the same is presumably true for permafrost-released CO2 and CH4. So can you at least comment on how the split into these*
30 *components affects TCRE and ZEC? (i.e. if all the carbon is released as CO2 it has a longer lived impact, but if it is released as CH4 it has a more immediate, but shorter lived, impact?)*

AUTHORS' RESPONSE:

The lifetime of both forcing agents (longer-lived for $CO_2$ short-lived for $CH_4$) is taken into account and handled internally by FaIR, opposed to treating their forcing timescales equally outside of FaIR (as referred to in line 52–53 of
35 the original manuscript). The respective contributions of $CO_2$ and $CH_4$ emissions is considered before they are fed into FaIR each annual time step. Since PerCX is calibrated using $CO_2$ in units of carbon and $CH_4$ in units of the carbon equivalent, the respective carbon emissions in PerCX are initially considered in units of carbon.

R1C3: *REVIEWER'S COMMENT:*
40 *Probably worth making clear the definition here of "permafrost" as it means different things to different people. You are (quite reasonably) referring to the response of currently-frozen carbon in the soil. A wider definition for example might in-clude changes in vegetation dynamics on frozen ground (warming/thawing allows forest expansion) – e.g. Pugh et al find this can be as large, and is a long term sink: https://agupubs.onlinelibrary.wiley.com/doi/full/10.1029/2018EF000935. But this is not what you cover. Somewhere inbetween these things, there is the impact of thawed nitrogen on en-*
45 *hancing vegetation growth – this is an interaction term between siol BGC and vegetation (see eg Burke et al: https://www.mdpi.com/2504-3129/3/2/23), but again I think this is explicitly not what you cover?*

AUTHORS' RESPONSE:

This assumption is (mostly) correct. Here, we refer to the response of currently-frozen carbon in the soil. PerCX itself does not account for further (localized) climate feedbacks, such as from vegetation interactions or its changing
50 distribution, nitrogen fertilization, or nonlinear dynamics of biogeochemical and biogeophysical processes. Hence, these modulating feedbacks are not (currently) included in PerCX owing to its simplicity, effectively emulating (the range of) responses of more complex climate models through its constraining process. However, implicitly, the responses of the more complex models likely include some of these processes (although ambiguous) that alter their positive and negative contributions to the existing permafrost carbon loss estimates used to calibrate PerCX, and are potentially
55 also a reason for the large range of possible outcomes in the reported estimates in AR6. In conclusion, PerCX does not (yet) account for any of these additional processes explicitly. We added a clarifying sentence regarding the scope of processes considered here to the end of the Introduction. Further, we want to note that individual model fits are

possible with PerCX (and we have done so in testing with outputs from the JULES land surface model), which opens the possibility for future explorations of the emulation of combined contributions of model specific processes relevant

60     for the permafrost carbon feedback.

R1C4: *REVIEWER'S COMMENT:*

*You say the results that can be applied as scaling to existing results (line 122). This is just for TCRE right? Whereas for ZEC it would be an addition rather than a scaling?*

65     AUTHORS' RESPONSE:

That is correct. We have adjusted the wording around this in the Conclusions section for clarity.

**Reviewer #2 (Andrew MacDougall)**

R2C1: *REVIEWER'S COMMENT:*

70    *While the functional form of PerCX appears like it should roughly capture the permafrost carbon feedback, PerCX is never tested against existing models. Unfortunately none of the existing permafrost MIPs provide the parameters needed to test PerCX against a suite of models. However the CO2 component of PerCX could be compared to the results from MacDougall 2021. These results are publicly available at: https://doi.org/10.5683/SP2/I75BZ0*

   *While it is sub-ideal to test PerCX against only a single other model it is better than testing it against no models.*

75    AUTHORS' RESPONSE:

   As suggested, we test PerCX (FaIR-PCF) against the UVic intermediate complexity Earth System Model output from MacDougall (2021). The comparison of modeling results is shown and further discussed in the Appendix A, which we refer to with an additional sentence in Section 2. Here, we also detail the necessary steps undertaken to replicate the *esm-1pct-brch-1000PgC* (A1) ZEC experiment with PerCX that allows a consistent comparison between the two

80    modelling exercises.

R2C2: *REVIEWER'S COMMENT:*

   *Line 11: Delete 'carbon'. While we usually give ZEC values from CO2 only ZEC, all forcing ZEC is a well established concept.*

85    AUTHORS' RESPONSE:

   We assume that this comment is referring to the occurrence of 'carbon' in line 13, where it is used in the explanation of what is meant by the Zero Emission Commitment (ZEC). We agree with this change, given that ZEC (or more specifically 'effective ZEC') refers to ZEC including all non-$CO_2$ forcings, and have therefore taken it out.

90 R2C3: *REVIEWER'S COMMENT:*

   *Line 48: Cp(0) is a constant, so probably should not be shown in functional form.*

   AUTHORS' RESPONSE:

   Indeed, while $C_p(0)$ is treated as a constant within this modeling exercise, it is the initial state of an otherwise (i.e., under warming) changing variable carbon pool $C_p(t)$, which in itself is not constant. $C_p(0)$ is also contextualized by

95    $C_p(t)$ and adding the former as a new constant would add a degree of freedom that does not actually exist. Further, understanding of $C_p(0)$ is aided by the first sentence following Equations 2 and 3. We therefore would prefer to keep its notation as is.

R2C4: *REVIEWER'S COMMENT:*

*Line 90: The TCRE values from MacDougall and Friedlingstein (2015) is an effective-TCRE value, not a CO2 only TCRE value, as it is derived from all forcing RCP simulations. The difference between effective-TCRE and TCRE was not well established when the paper was published. eTCRE values tend to be higher than TCRE values.*

AUTHORS' RESPONSE:

We appreciate this comment. We have added a clarifying sentence to this paragraph that clarifies this difference. Further, due to the consequences of this difference to the uncertainty of inferring UVic's PCF, we have softened the statement at the end of this paragraph.

R2C5: *REVIEWER'S COMMENT:*

*Quotes in Latex are '' Using '' will give two end quotes*

AUTHORS' RESPONSE:

We have addressed this technicality in the appropriate places in the text and expect that potential further misalignments are going to be corrected during the typesetting process.

**References**

115  Canadell, J., Monteiro, P., Costa, M., da Cunha, L. C., Cox, P., Eliseev, A., Henson, S., Ishii, M., Jaccard, S., Koven, C., Lohila, A., Patra, P., Piao, S., Rogelj, J., Syampungani, S., Zaehle, S., , and Zickfeld, K.: Climate Change 2021: The Physical Science Basis. Contribution of Working Group I to the Sixth Assessment Report of the Intergovernmental Panel on Climate Change, chap. Global Carbon and other Biogeochemical Cycles and Feedbacks, pp. 673–816, Cambridge University Press, Cambridge, United Kingdom and New York, NY, USA, 2021.